# Rapid and sound assessment of well-being within a multi-dimensional approach: The *Well-being Numerical Rating Scales* (WB-NRSs)

**Andrea Bonacchi**[1,2]☉**, Francesca Chiesi**[3]☉*****, Chloe Lau**[4]**, Georgia Marunic**[5]**, Donald H. Saklofske**[4]**, Fabio Marra**[6]**, Guido Miccinesi**[1]

**1** Clinical Epidemiology Unit, Oncological Network, Prevention and Research Institute -ISPRO, Florence, Italy, **2** Centro Studi e Ricerca Syntesis, Florence, Italy, **3** Department of Neuroscience, Psychology, Drug, and Child's Health (NEUROFARBA), University of Florence, Florence, Italy, **4** Department of Psychology, Western University, London, Canada, **5** School of Psychology, University of Florence, Florence, Italy, **6** Experimental and Clinical Medicine Department, University of Florence, Florence, Italy

☉ These authors contributed equally to this work.
* francesca.chiesi@unifi.it

**Data Availability Statement:** All relevant data are within the manuscript and its Supporting Information files.

## Abstract

The assessment of well-being remains an important topic for many disciplines including medical, psychological, social, educational, and economic fields. The present study assesses the reliability and validity of a five-item instrument for evaluating physical, psychological, spiritual, relational, and general well-being. This measure uniquely utilizes a segmented numeric version of the visual analog scale in which a respondent selects a whole number that best reflects the intensity of the investigated characteristic. In study one, 939 clinical (i.e., diagnosed with cancer and liver disease with cirrhosis) and non-clinical (i.e., undergraduate students and their family and acquaintances) participants between the ages of 18 to 87 years ($M$ = 47.20 years, $SD$ = 19.62, 54% males) were recruited. Results showed items have strong discriminant ability and the spread of threshold parameters attests to the appropriateness of the response categories. Moreover, convergent and discriminant validity were found with other self-report measures (e.g., depression, anxiety, optimism, well-being) and the measure showed responsiveness to two separate interventions for clinical populations. In study two, 287 Canadian (ages ranged from 18 to 30 years; $M$ = 20.78, $SD$ = 3.32; 23% males) and 342 Italian undergraduate psychology students (age ranged from 18 to 29 years, $M$ = 21.21 years, $SD$ = 1.73, 38% males) were recruited to complete self-report questionnaires. IRT-based differential item functioning analyses provided evidence that the item properties were similar for the Italian and English versions of the scale. Additionally, the validity results obtained in study one were replicated and similar relationships between criterion variables were found when comparing the Italian- and the English-speaking samples. Overall, the current study provides evidence that the Italian and English versions of the WB-NRSs offer added value in research focused on well-being and in assessing well-being changes prompted by intervention programs.

**Funding:** The authors received no specific funding for this work.

**Competing interests:** The authors have declared that no competing interests exist.

## Introduction

Well-being remains an important area of study for many disciplines including medical, psychological, social, educational, and economic fields. This interest has gradually grown over the past 50 years, among scientists, healthcare professionals, politicians, and laypersons, and the number of self-report tools designed and developed specifically to measure subjective well-being has increased substantially [1, 2]. Currently, there are numerous distinct self-report instruments that range from several to more than three-hundred items (for a review [1]) that vary on the theoretical and practical conceptualizations of well-being, the number and classification of well-being dimensions assessed, and answer options (i.e., Likert-type scales, yes/no options, or pictorial items).

While numerous well-being theories have been developed over recent decades, only several have been influential in the design of self-report measures, including the WHO definition of health [3], Dieners' model of subjective well-being based on affective and cognitive components [4], and Maslow's description of hierarchically distributed basic human needs [5]. Other influential theories of well-being include Sen's capability approach in moral significance and values [6], Antonovsky's theory of salutogenenesis in resources to cope with stressors [7], and Ryff's six-factor model of psychological well-being including self-acceptance, positive relationships with others, personal mastery, autonomy, purpose and meaning in life, and personal growth and development [8].

In the current study, our theoretical approach is based on the assumption that personality characteristics or individual dispositions (i.e., resources such as sense of coherence [7], dispositional optimism [9], locus of control/sense of mastery [10]) allow us to more effectively meet our psychological, mental, social, and existential needs, and, as a consequence, to experience a greater personal degree of well-being across different domains. We incorporated several theoretical approaches to well-being in the short, yet comprehensive, scale we aimed to develop. First, our theoretical approach is based on the WHO definition of health as "*a state of complete physical, mental and social wellbeing, not merely the absence of disease*" [3]. This approach provides a multidimensional definition to well-being in measuring three core components: (1) *physical well-being*, consisting of a sense of pleasure that derives from meeting physical needs (e.g., rest, food, sex) and comfort that derives from the quality of bodily functioning and the feeling of having adequate energy to meet daily needs [1]; (2) *mental well-being*, including a personal experience of happiness and positive feelings regarding satisfaction of psychological needs (e.g., for positive change, fun and engaging activities, knowledge); (3) *social well-being*, reflecting how positively an individual is connected to others in the social community and the possibility to experience a sense of harmony and integration due to the gratification of social needs (e.g. need for appreciation, need to have a positive role in the community, need for support). Additionally, we deem relevant a fourth dimension: *spiritual well-being*, which consists of the sense of serenity and joy that derive from the gratification of existential needs (e.g., to live coherently with one's values and with the recognized meaning of life, to be satisfied with one's lived life and the goals achieved) and spiritual needs (e.g., for connection with something greater than oneself) [11]. Finally, we consider *general well-being* as the expression of the balance among these four dimensions that might result in more than their simple summation. This multidimensional perspective on well-being is graphically represented in Fig 1.

Among the various tools for measuring well-being, some have been developed in a multidimensional perspective, such as the *Biopsycosocialspiritual Inventory* (BIOPSSI) [12] and the *Mental Physical Spiritual Well-Being Scale* (MPS) [13]. However, these measures consist of a large number of items (e.g., the BIOPSSI has 41 items, the MPS has 30 items). While a multidimensional approach to well-being is comprehensive, there is the added value offered by a brief

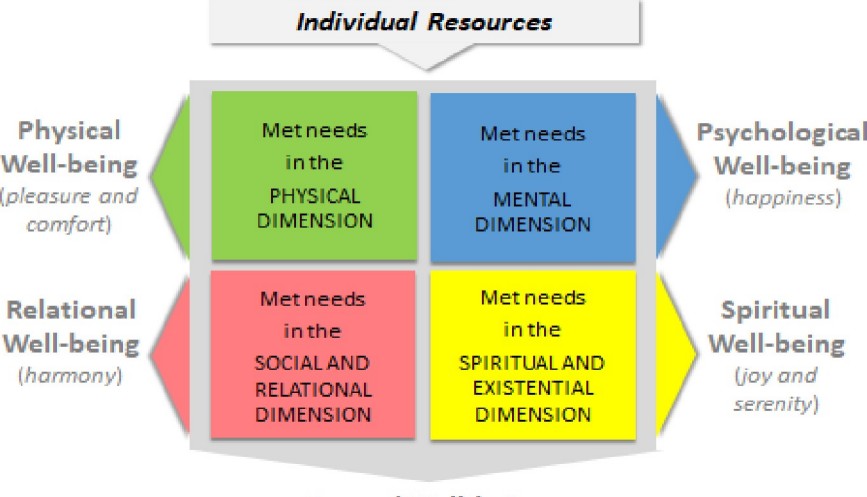

**Fig 1. Multidimensional model of well-being.**

scale. Indeed, an effective, concise, and centered measure has numerous benefits in both research and clinical practice [14, 15]. Respondents are less likely to experience fatigue, boredom, loss of interest or perceive the questionnaires as time-demanding during administration (an aspect of particular relevance when administering the scale to patients, for example). Moreover, it is more appropriate for large, multivariate studies in which many tests and scales may require administration, and it can be employed in rapid surveys or to test quickly pre-post changes after an intervention program.

First of all, we posited that the aim of the present study is to develop a tool to rapidly and accurately assess the aforementioned dimensions of well-being.

First of all, we posited that the aforementioned dimensions of well-being can be assessed using a numerical rating scale (NRS). The NRS is a segmented numeric version of the visual analog scale in which a respondent selects a whole number that best reflects the intensity of the investigated characteristic (usually pain or a symptom). In most cases, it is represented by a horizontal line or bar anchored by numbers corresponding respectively to "none" and "totally". This type of scale has several merits for both the test-taker and test-user, such as high comprehensibility and ease of completion, short administration time (usually less than one minute), verbal (therefore also by telephone) or graphical (usually by self-completion) administration options, and simplicity of administration and scoring [16, 17]. Moreover, the methodology has been deemed reliable and valid [18, 19] and requires minimal language translation, which supports the use of the NRS across time, cultures, and languages [17]. Hence, relying on the aforementioned theoretical approach (graphically represented in Fig 1), we developed the *Well-being Numerical Rating Scales* (WB-NRSs) consisting of five NRSs to rate physical, psychological, spiritual, relational, and general well-being. We provided evidence that the WB-NRSs allow for a valid assessment of well-being and its components. The psychometric properties were tested in two different studies.

In the first study, each item was examined through the application of item response theory (IRT) [20, 21]. IRT is a parametric statistical modeling procedure which involves fitting a hypothetical model to data starting from the assumption that the probability of an item response depends on the respondent's characteristic and the item characteristics. Specifically, the IRT model provides parameters that enable an evaluation of how well an item performs in

measuring the construct targeted by the item and the appropriateness of the response categories [22]. Along with the item properties, we also tested the validity of the scale. Specifically, we used measures of general, psychological, relational, and physical well-being to support that each item assesses a specific dimension of well-being and that overall well-being includes all these components. Then, we investigated the relationships of the well-being dimensions with several psychological constructs (i.e., dispositional optimism, sense of coherence, sense of mastery, stress, anxiety, and depression) seeking to replicate the nomological net documented in the literature (for a review [1]). Finally, we investigated the responsiveness of the scale. We hypothesized that each item would detect changes in well-being after specific intervention strategies in very different domains. One intervention was in the health domain: cancer patients were invited to attended live classical music concerts during the hospitalization period to reduce their distress [23, 24]. The other intervention was in the art domain: participants were invited to an immersion itinerary developed by Gianni Falvo (2014) that focused on the observer's capacity to enter into contact with the work of art through different sensory channels [25, 26]. Both the interventions were original strategies (using music and artistic features) specifically planned for improving participants well-being. Thus, the effectiveness of the proposed measure in detecting such a change can be considered an important feature of the instrument.

In the second study, to increase the generalizability of the WB-NRSs and its usability, the metrical equivalence of the Italian and the English version of the WB-NRSs was investigated to determine if the current assessment tool, originally developed in Italian, might be considered metrically invariant once translated into English and administered to English-speaking people. An IRT-based Differential Item Functioning (DIF) analysis was used to investigate whether the WB-NRSs are differently understood or interpreted depending on the respondent's language. DIF analysis tests the assumption that the likelihood of a response option endorsement should be equal across subgroups, which are matched on the trait measured (i.e., responses should be the same for Italian- or English-speaking people having the same well-being level). Additionally, in this second study, we aimed to confirm some of the relationships observed in the previous validity testing (Study 1) and to replicate the observed patterns of relationships in the English sample.

## Study 1

### Methods

**Participants.** A total of 939 participants between the ages of 18 to 87 years ($M$ = 47.20, $SD$ = 19.62, 54% males) were recruited for the study. The sample includes both clinical and non-clinical participants. The diversity of the sample allowed us to determine the generalizability of findings in the general population by age and gender. The sampling of clinical participants allowed for testing of the effectiveness of the scale in the health domain. Indeed, the scale was developed to be used in these settings in order to evaluate well-being in a treatment setting.

Clinical participants were diagnosed with cancer and liver disease with cirrhosis. Patients were taking part in a wider survey of patients' unmet needs and complementary therapies. Participation was voluntary and offered to all patients visiting outpatient clinics or admitted to oncology wards. The percentage of patients who volunteered to take part in the research in the different medical units ranged from 71.0% and 95.4%. After applying some exclusion criteria (age under 18 or over 90 years, cognitive impairment, comorbid psychotic illness, learning disability, severe symptoms due to illness or side effects of therapy that precluded the possibility to fill in questionnaires individually), the sample was composed of 366 patients (ages ranged

from 35 to 89 years; $M$ = 63.67, $SD$ = 10.85; 62% males). Approval was obtained from the local ethics review board of the academic healthcare setting (Comitato Etico per la Sperimentazione Clinica dei medicinali, n. 46498/67; Comitato Etico Locale Azienda Ospedaliero-Universitaria Careggi, n. 10574_oss).

For non-clinical cases ($N$ = 573, age ranged from 18 to 84 years, $M$ = 37.00 years, $SD$ = 16.67, 49% males), sampling was based on the "snowball" method [27]. Undergraduate psychology students were invited to participate in the study and were also encouraged to recruit their acquaintances and relatives to participate. Participation in the study was voluntary and students received a credit towards their psychology course. The study was approved by the university's local institutional review board (Commissione Etica per la Ricerca dell'Università degli Studi di Firenze, n. 31—prot. 127556).

Written informed consent was obtained from all participants.

**Measures.**   *Well-being Numerical Rating Scales* (WB-NRSs). Item generation relied upon literature reviews and the aforementioned four-dimensional theoretical model of well-being (see also Fig 1). Health professionals' (physicians, nurses, and psychologists) also evaluated the content validity of the items based on their clinical experience. Hence, the five items were developed and initially administered to a small convenience sample ($n$ = 15) to check for understanding. All the items were clearly understood, and no overlapping or misunderstandings were detected. Respondents are asked to rate their physical (WB-NRS 1), psychological (WB-NRS 2), relational (WB-NRS 3), spiritual (WB-NRS 4), and general well-being (WB-NRS 5) using a 1-to-10-point scale with 1 indicating a state of "absolute distress" and 10 a state of "complete well-being."

*Hospital Anxiety and Depression Scale* (HADS) [28], Italian version [29]. The HADS consists of 14 items that screen for symptoms related to depressed mood and anxious symptoms with higher scores representing greater distress. The questionnaire consists of two factors of anxiety and depression: HADS-A (7 items) measures anxiety (e.g.,"I feel tense or 'wound up'") and HADS-D (7 items) measures depression (e.g., "I feel as if I am slowed down"). In the current sample, both subscales showed adequate internal consistency (HADS-A: $\omega$ = .77; HADS-D: $\omega$ = .77).

*Short Form 12 items Health Survey* (SF-12) [30]. The SF-12 is an abbreviated version of the *Short Form 36 items Health Survey* (SF-36) which assesses the individual's subjective perception of the concept of health, understood as biopsychosocial well-being [31]. The SF-12 focuses on different aspects related to physical health (e.g., "Have you had any of the following problems, e.g., climbing flights of stairs, with your work or other regular daily activities as a result of your physical health?") and mental health (e.g., "Have you felt down-hearted and blue?"). The total scores represent a physical health index (PSF-12) and a mental health index (MSF-12). Lower scores represent greater levels of disability. In the current sample, both subscales have an adequate internal consistency (PSF-12: $\omega$ = .80; MSF-12: $\omega$ = .85).

*Psychological Well-being Scales* (PWBS) [32], Italian version [33]. The PWBS contains eighty-four Likert-type items using a six-point scale ranging from "strongly disagree" to "strongly agree". Six dimensions are assessed including self-acceptance, autonomy, environmental mastery, personal growth, purpose in life, and positive relations. *Self-acceptance* (14 items) is the attitude toward oneself and one's past life (e.g., "When I look at the story of my life, I am pleased with how things have turned out"). *Autonomy* (14 items) describes the sense of independence and acting out of the bounds of cultural norms (e.g., "I have confidence in my opinions, even if they are contrary to the general consensus"). *Environmental Mastery* (14 items) measures the ability to manage life and one's surroundings (e.g., "In general, I feel I am in charge of the situation in which I live"). *Personal Growth* (14 items) assesses the inclination to pursuit of a continued personal development (e.g., "I think it is important to have new

experiences that challenge how you think about yourself and the world"). *Purpose in Life* (14 items) describes having life goals and beliefs that one's life is meaningful (e.g., "Some people wander aimlessly through life, but I am not one of them"), *Positive Relations* (14 items) deals with the quality of the relationships with others (e.g., "People would describe me as a giving person, willing to share my time with others"). In the current sample, the internal consistency of each subscale is good ($ω = .90$ for *Self-acceptance*; $ω = .82$ for *Autonomy*; $ω = .86$ for *Environmental Mastery*; $ω = .82$ for *Personal Growth*; $ω = .86$ for *Purpose in Life*; $ω = .87$ for *Positive Relations*).

*World Health Organization Well-Being Index* (WHO-5) [34, 35]. The WHO-5 is a five-item measure used to measure subjective psychological well-being (e.g., "I have felt cheerful in good spirits"). The measure has been translated to over 30 languages and respondents rate each statement according to a six-point Likert-style scale from "at no time" to "all of the time." In the current sample, McDonald's $ω$ was .85, indicating good reliability.

*Perceived Stress Scale—4 item* (PSS-4) [36]. The PSS-4 is a widely used psychological instrument for measuring the degree to which situations in one's life are appraised as stressful (e.g., "In the last month, how often have you felt difficulties were piling up so high that you could not overcome them?"). The construct was measured with four items using a five-point Likert scale from "never" to "very often". Higher scores indicate greater stress. In the current sample, McDonald's $ω$ was .80, indicating adequate reliability.

*Pearlin-Schooler Mastery Scale* (PSMS) [10]. The PSMS consists of seven items on a five-point Likert scale (from "strongly disagree" to "strongly agree) measuring sense of mastery, defined as the extent to which one regards one's life as being under one's control in contrast to being fatalistically ruled (e.g., "I can do just about anything I really set my mind to"). Higher scores represent stronger sense of mastery. The scale has a good internal consistency in the current sample ($ω = .83$).

*Life Orientation Test—Revised* (LOT-R) [9], Italian version [37]. The LOT-R measures dispositional optimism defined as a generalized expectancy of positive future outcomes. It consists of six items (e.g., "In uncertain times, I usually expect the best") and four filler items answered on a five-point Likert scale from "strongly disagree" to "strongly agree". Higher scores indicate higher dispositional optimism. The scale has a good internal consistency in the current sample (McDonald's $ω = .80$).

*Sense of Coherence Scale—13 item* (SOC-13) [7], Italian version [38]. The SOC-13 consists of items scaled along a seven-point semantic differential with two anchoring phrases at both extremes. Higher scores indicate higher level of SOC. In the current sample the scale has an adequate internal consistency ($ω = .84$).

*Sense of Coherence—3 items* (SOC-3) [39], Italian version [40], is an ultra-brief scale that consists of three items matching the three comprehensibility; manageability, and meaningfulness components of SOC as defined by Antonovsky [7]. The response format includes three options: no, sometimes, and yes. Higher scores indicate higher level of SOC.

**Design and procedure.** A descriptive observational research design was adopted in this cross-sectional study with the exception of the test-retest component for responsiveness testing (repeated measures design). Data were collected from January 2017 to April 2019.

The WB-NRSs was presented to the sample participants as part of the questionnaire including other measures employed for validity testing. Specifically, participants were randomly asked to complete one of four different questionnaires developed for research and clinical purposes. Clinical participants ($N = 366$) were administered a paper-and-pencil questionnaire that included the WB-NRSs and HADS. A subsample ($N = 110$) also completed the SF-12. Among non-clinical participants, one group (Sample 1: $N = 231$) were given a paper-and-pencil questionnaire that included the WB-NRSs, the WHO-5, the PSS-4, the LOT-R, the PSMS, the

PWBS, and the SOC-13. The remaining participants (Sample 2: *N* = 342) were administered an online questionnaire (using GoogleForms) that included the WB-NRSs, the WHO-5, the PSS-4, the LOT-R, the PSMS, and the SOC-3. Depending on the length of the questionnaire, administration time ranged from 15 to 45 minutes.

To investigate responsivity, we employed a sub-group of the clinical sample (*N* = 256) who took part to the "Donatori di Musica" [Music Givers] interventions (described in detail in [23, 24]). The psycho-oncologists of the units proposed the study to inpatients assuring them that participation was free and on a voluntary basis and that non-adherence did not alter care received by the staff of the ward. Each patient was interviewed by psycho-oncologists before (Time 1) and after (Time 2) a Music Givers live concert that took place once a week in a dedicated space within the department of oncology. Along with the HADS, they were asked to fill the WB-NRSs. Pre- and post-data were collected for 245 inpatients.

Responsivity was also tested by administering the WB-NRSs at the beginning (Time 1) and at the end (Time 2) of an aesthetic experience in the Chapel of the Magi in Palazzo Medici Riccardi, in Florence (for a detailed description [25, 26]). The participants (*N* = 23, *M* = 52.22 years, *SD* = 8.92, 48% males) were proposed some workshops, presented during two consecutive days, where the experience was amplified moving from the direct observation of the fresco to insight into specific details which are filmed and reproduced in 4K video format, adding cognitive information (e.g., the pictorial technique, the description of details) and emotional-sensory stimulus (e.g., music tracks).

**Analysis strategy.** Prior to conducting the analyses, we examined the missing values in the data. Listwise deletion was used when one or more answers of the WB-NRSs were missing. For the other scales, listwise deletion was used when a case had more than 10% of missing answers [41]. Otherwise, the case item mean was used to replace the missing value.

We performed IRT analyses using IRTPRO 4.0 [42]. IRT provides an explicit measurement model of expressing association between individual differences in observable manifestation of a trait and the underlying latent trait θ [20, 43, 44]. Based on the WB-NRSs response format, Samejima's graded response model (GRM), one of the most used models for graded polytomous data, was chosen [45].

Preceding the IRT analysis, we computed the item descriptive statistics on SPSS version 26.0 because the variability in item responses is a prerequisite for IRT parameter estimation. Similarly, item-total correlations were computed because their variability indicates the adequacy of a discrimination parameter estimation in IRT. Additionally, a fundamental assumption of the GRM is unidimensionality. Hence, a parallel analysis based on minimum rank factor analysis and an unrestricted factor analysis using a Robust Maximum Likelihood estimation method were conducted on FACTOR [46]. The Nonnormed Fit Index (NNFI), Comparative Fit Index (CFI), and Root Mean Square Error of Approximation (RMSEA) were employed to evaluate the goodness-of-fit. As recommended by Byrne, NNFI and CFI $\geq$ .95 along with RMSEA $\leq$ .08 would suggest good model fit [47]. Finally, another prerequisite for IRT is to detect local dependence (LD). The absence of LD is important for unidimensional IRT modelling because the item parameter estimates are adequate only if there is not an excess of covariation among item responses that is not accounted for by a unidimensional IRT model. The LD was assessed using the diagnostic $\chi^2$LD statistic that is approximately distributed as standardized $\chi^2$ [48]. Given this approximation, as a rule of thumb, values of 10 or greater indicate the presence of LD.

Subsequently, the GRM was applied. The model's goodness of fit was evaluated using the $M_2$ statistic and the associated root mean square error of the approximation (RMSEA) value. As the $M_2$ statistic, similar to other $\chi^2$statistics, is generally unrealistic because some error will be present in any strong parametric model, the RMSEA provides a metric for model errors [49]. RMSEA values of .05 or less indicate a good fit. Then, using the marginal maximum likelihood estimation,

we obtained the item parameters, i.e., the discrimination ($a$) and location ($b_i$) parameters. The $a$ parameter describes the ability of an item to discriminate among people with different levels of the underlying trait (i.e., the higher $a$ is, the higher the item's ability to differentiate between people with different levels of well-being). The $b$ parameters, also called category thresholds, represent an item's sensitivity in differentiating among the various levels of the target trait (i.e., if $b$ values are evenly spaced along the trait, the item categories provide a better differentiation in measuring well-being). Ten parameters were estimated for each item, including one $a$ parameter and nine $b$ threshold parameters (that are equal to the number of response options minus one). Following Baker and Kim, $a$ values below 0.24 are considered very low, 0.25 to 0.64 are considered low, 0.65 to 1.34 are considered moderate, 1.35 to 1.69 are considered high, and more than 1.7 are very high [50]. The $b$ parameters should be evenly spaced along the trait to provide a differentiation and variability along the latent continuum when measuring the trait. The graphical representations of both item parameters are called item characteristic curve (ICC). Finally, the Item Information Function (IIF), graphically represented by the Item Information Curve (IIC), describes the amount of information that a particular item provides across the entire continuum of the latent construct, and it depends on both $a$ and $b$ parameters. Thus, the IICs of the five items were individually examined to investigate their precision in assessing each dimension.

Bayesian analyses were used to evaluate the scale validity and responsiveness. Jeffreys' Bayes Factor described the observed data using a priori and posterior distribution [51], which allowed quantification of evidence in favor of the alternative and null hypothesis [52]. Bayes Factors for evidence of alternative hypotheses is presented as an easy-to-interpret odds ratio that represents the magnitude of the difference: 1–3 as weak, 3–10 as substantial, 10–30 as strong, 30–100 as very strong, and >100 as decisive [53]. All the Bayesian tests were performed under a default uniform prior [54] using JASP 0.14 [55].

Bayesian correlation tests were conducted to test the validity of the WB-NRSs. Convergent and discriminant validities are two aspects of construct validity. Convergent validity refers to the extent to which assessment of the same or similar traits intercorrelate with one another. Discriminant validity involves demonstrating that a measure does not correlate too strongly with measures they are not intended to. Thus, we expected high correlations ($r \geq 0.55$) between the WB-NRSs and the scales that measure very similar well-being components (e.g., WHO-5 and WB-NRS 2; SF-12 physical health index and WB-NRS 1; PWBS *Positive Relations* scale and WB-NRS 3), and lower correlations ($r \leq 0.45$) between different aspects of well-being (WHO-5 and WB-NRS 1; SF-12 physical health index and WB-NRS 2; PWBS *Positive Relations* scale and WB-NRS 1). Additionally, the degree to which the scores correlated with external theoretically connected variables, provide evidence of construct and criterion validity. Hence, medium and negative in size ($-0.50 < r < -0.40$) correlation were expected for stress, depression, and anxiety. Finally, significant but low to medium in size correlations ($0.25 < r < 0.45$) were expected between WB-NRSs and personality characteristics or individual disposition that are deemed to promote well-being (such as, optimism, sense of coherence, and sense of mastery).

Bayesian dependent sample *t*-tests were used to evaluate the scale ability to detect change (i.e., responsiveness). Since we hypothesized an increase in well-being through the course of interventions, we tested a one-tail hypothesis (i.e., the score at Time 1 is lower compared to the score at Time 2).

## Results and discussion

Minimal data were missing across all variables. For the WB-NRSs, the missing values remained under 2% of the total cases in the sample. Thus, listwise deletion was used and 14 cases were excluded.

**Prerequisites for IRT analysis.** *Item descriptives*. All indices are reported in Table 1. Descriptives of each item showed that respondents used all the possible scale options and answers were well distributed along the range of the numerical scale (standard deviations were all around 2). Means (values were around 6.5) and skewness indices (all negative in sign) indicated that the frequencies were slightly fewer in the lower part of the response scale (i.e., answers were more concentrated on the higher values). Nonetheless, all the skewness and kurtosis indices ranged within -1, suggesting that distributions were similar to a normal distribution. Item–total correlation values ranged from .59 - .81 suggesting that the variability in item discrimination parameters would warrant inclusion in the IRT calibration.

*Unidimensionality*. The parallel analysis clearly advised a one-factor structure. The model demonstrated a very good fit ($\chi^2/df$ = 3.70, CFI = .99, NNFI = .99, RMSEA = .053 [95% CI: .049 -.057]). The variance explained by the model was 65% and all factor loadings were higher than 0.65 (Table 1).

*Local Dependence*. None of the LD statistics were greater than 10 (the maximum value was 9.3), indicating the absence of covariations between couples of items that are not accounted for by a unidimensional model.

Overall, these results showed that IRT analysis can be performed to test the psychometric properties of the items.

**Item response theory analysis.** *Item parameters*. The fit for the GRM was adequate ($M_2$ = 2398.40, $df$ = 805, $p < .001$; RMSEA = .05). Thus, we examined the parameter estimates. Following Baker's cut-offs [56], four items had a very high discriminative power with Item 2 and Item 5 showing the highest values (Table 2). Hence, all items can adequately distinguish between individuals with different levels of physical, psychological, relational, spiritual, general well-being and, in particular, psychological and general well-being items are the best performing ones.

The nine $b$ threshold parameters were evenly spaced. The $b_1$ and $b_2$ were from about 3.00 to 2.00 $SD$s below the mean trait (fixed at 0.00, SD = 1.00, by default). The $b_3$, $b_4$, and $b_5$ were from about 2.00 to 0.50 $SD$s below the mean, while $b_6$ and $b_7$ were around the mean, and, finally, $b_8$ and $b_9$ from 1.00 to 2.00 $SD$s above the mean (Table 2). These values indicated that all items showed an adequate increase of the latent trait level at each subsequent response option (i.e., the items perform well in measuring a large spectrum of the underlying construct). Nonetheless, because ideally 3 SDs below and above the mean trait should be covered, the five items fail to cover adequately the highest levels of the trait.

Fig 2 shows the ICCs for each item used in IRT to provide visual information regarding the item characteristics. The graphs indicated that there was a moderate separation in the response options and that the curves of each response option were distributed across the trait range (except for the highest levels). Whereas all the items response curves are partially overshadowed by the neighbor categories, each category shows a specific level of trait for which the

**Table 1. Descriptives, item-total correlations (with item deleted), and factor loadings of the *Well-being Numerical Rating Scales* (WB-NRSs).**

|  | *Range* | *Mean* | *Standard Deviation* | *Skewness* | *Kurtosis* | *Item-Total correlation* | *Factor Loading* |
|---|---|---|---|---|---|---|---|
| WB-NRSs |  |  |  |  |  |  |  |
| 1 –Physical WB | 1–10 | 6.49 | 2.06 | -0.51 | -0.22 | 0.59 | 0.64 |
| 2 –Psychological WB | 1–10 | 6.54 | 2.14 | -0.46 | -0.33 | 0.77 | 0.86 |
| 3 –Relational WB | 1–10 | 7.27 | 1.96 | -0.75 | 0.34 | 0.63 | 0.68 |
| 4 –Spiritual WB | 1–10 | 6.88 | 2.25 | -0.56 | -0.32 | 0.61 | 0.65 |
| 5 –General WB | 1–10 | 6.75 | 1.90 | -0.63 | 0.20 | 0.81 | 0.89 |

**Table 2. Item response theory parameters of the *Well-being Numerical Rating Scales* (WB-NRSs).**

| Item | a (SE) | $b_1$ (SE) | $b_2$ (SE) | $b_3$ (SE) | $b_4$ (SE) | $b_5$ (SE) | $b_6$ (SE) | $b_7$ (SE) | $b_8$ (SE) | $b_9$ (SE) |
|---|---|---|---|---|---|---|---|---|---|---|
| WB-NRS 1 | 1.74 (0.10) | -3.15 (0.19) | -2.52 (0.14) | -1.88 (0.10) | -1.33 (0.08) | -0.66 (0.06) | -0.13 (0.05) | 0.54 (0.06) | 1.36 (0.09) | 2.41 (0.14) |
| WB-NRS 2 | 3.23 (0.18) | -2.53 (0.12) | -1.99 (0.09) | -1.49 (0.07) | -1.08 (0.05) | -0.57 (0.05) | -0.16 (0.04) | 0.40 (0.05) | 1.00 (0.06) | 1.70 (0.08) |
| WB-NRS 3 | 1.86 (0.10) | -3.62 (0.26) | -2.79 (0.16) | -2.26 (0.12) | -1.79 (0.09) | -1.23 (0.07) | -0.70 (0.06) | 0.00 (0.05) | 0.85 (0.0) | 1.57 (0.09) |
| WB-NRS 4 | 1.76 (0.10) | -3.14 (0.19) | -2.41 (0.13) | -1.85 (0.10) | -1.42 (0.08) | -0.83 (0.06) | -0.35 (0.05) | 0.21 (0.06) | 0.88 (0.07) | 1.60 (0.10) |
| WB-NRS 5 | 4.04 (0.28) | -2.60 (0.13) | -2.25 (0.10) | -1.70 (0.07) | -1.24 (0.05) | -0.75 (0.04) | -0.29 (0.04) | 0.34 (0.05) | 1.00 (0.06) | 1.80 (0.09) |

*Note*: *a* = discrimination parameter; *b* = threshold parameter; *SE* = standard error.

probability to choose it was higher. For example, low trait respondents (around –2.00) have a higher probability of endorsing option 3 or 4, and high trait respondents (around 1.00) have a higher probability of endorsing option 7 or 8. In particular, the response curves of Item 5 were very well spaced with high picks around the corresponding response option.

*Item Information.* The GRM was applied to estimate the IIF. The IIF describes the amount of information that a particular item provides across the entire continuum of the latent construct. As shown in Fig 2, items showed different information characteristics. Item 1 conveyed a moderate amount of information, but constantly along the entire trait (within the range from 3.00 SDs below to 3.00 SD above the mean trait). Item 3 and Item 4 conveyed a similar amount of information, but we observe a sharp drop in the range from about 2.00 SDs to 3.00 SDs above the mean. Item 2 conveyed a large amount of information from -3.00 to 2.00 SDs,

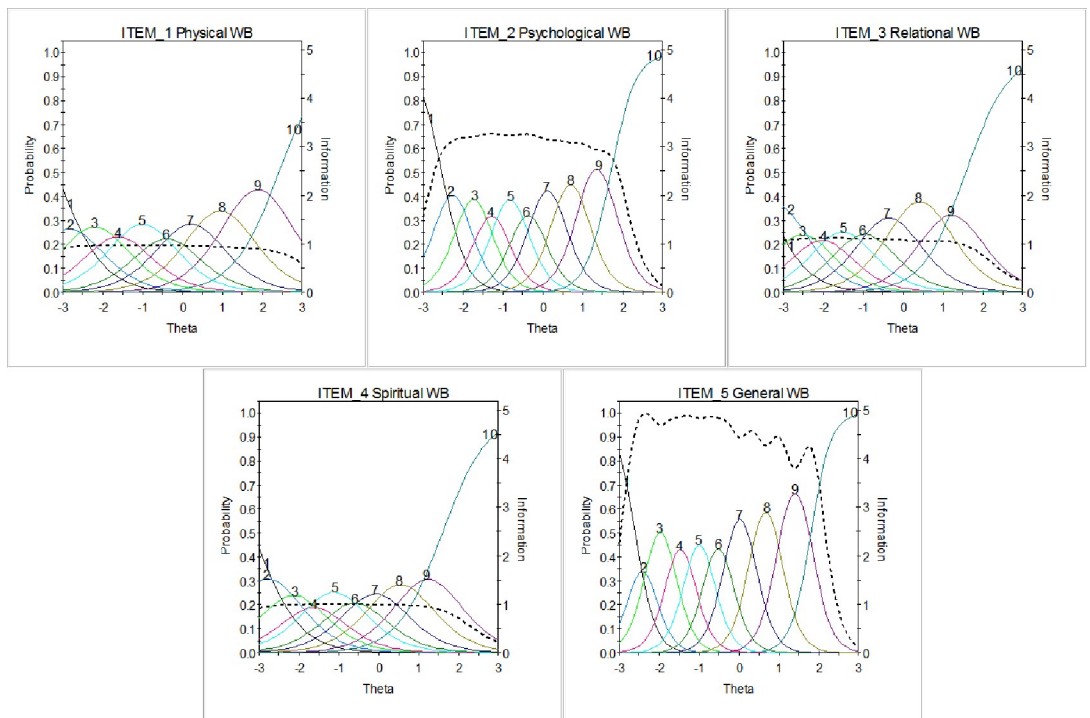

**Fig 2. Item Characteristic Curve (ICC) and Item Information Function (IIF) of the 5 items of the *Well-being Numerical Rating Scales* (WB-NRSs).** Latent trait (Theta) is shown on the horizontal axis, the probability of endorsing a response option (solid line) is shown on the left vertical axis, and the amount of information (dotted line) yielded by the item at each trait level is shown on the right vertical axis.

but a similar drop can be observed for trait levels higher than 2.00 SDs. Finally, Item 5 was greatly informative for almost the same range of the trait. Overall, measures of physical, social, and spiritual well-being are less precise, while psychological and general well-being are highly reliable, whereas less precise for the highest levels of the trait.

**Validity.** Bayesian correlation tests were computed to test the validity of the WB-NRSs. As a preliminary step to analyses, we checked the normality of the score distributions. All the skewness and kurtosis indices ranged within -1 and 1, suggesting that distributions were similar to a normal distribution.

As expected, evidence of convergent and discriminant validity was found (Tables 3–5). High correlations were observed between comparable dimensions of well-being, such as the WB-NRS 1 and the physical health index derived from the SF-12 ($r = 0.59$ in the clinical sample), between the WB-NRS 2 and the WHO-5 ($r = 0.62$ and $r = 0.64$ in the non-clinical Sample 1 and Sample 2, respectively) as well as with the mental health index derived from the SF-12 ($r = 0.64$ in the clinical sample), and the WB-NRS 3 and the *Positive Relations* scale of the PWBS ($r = 0.57$ in Sample 1). By comparison, low to medium correlations were observed between different dimensions of well-being, such as the WB-NRS 2 and the physical health index derived from the SF-12 ($0.16 < r < 0.20$ in the clinical sample), the WB-NRS 1 and the WHO-5 ($r = 0.39$ and $r = 0.45$, respectively in the non-clinical Sample 1 and Sample 2), and the WB-NRS 1 and the *Positive Relations* scale of the PWBS ($r = 0.09$ in Sample 1). The WB-NRS 4 was moderately correlated with the SF-12 mental health index ($r = 0.53$ in the clinical sample), the WHO-5 ($r = 0.51$ and $r = 0.41$ in the non-clinical Sample 1 and Sample 2, respectively), but low correlations were obtained for the physical dimension (i.e., $r = 0.20$ with the SF-12 physical health index in the clinical sample) and the relational dimension (i.e., $r = 0.21$ with PWBS *Positive Relations* in Sample 1). Finally, consistent with the proposed theoretical model, the WB-NRS 5 was strongly correlated with the SF-12 mental health index ($r = 0.70$ in the clinical sample), the WHO-5 ($r = 0.70$ and $r = 0.64$ in the non-clinical Sample 1 and Sample 2, respectively), but it also includes aspects related to the physical dimension (i.e., $r = 0.39$ with the SF-12 physical health index in the clinical sample) and the relational dimension (i.e., $r = 0.37$ with PWBS *Positive Relations* scale in Sample 1).

Additionally, we provided further validity evidence computing the correlations among the WB-NRSs and theoretically connected variables (Tables 3 and 4). Negative and medium to

**Table 3. Bivariate Bayesian correlates between the *Well-being Numerical Rating Scales* (WB-NRSs) and the other variables in the study in the clinical sample.**

|  | (1) | (2) | (3) | (4) | (5) | (6) | (7) | (9) | (10) |
|---|---|---|---|---|---|---|---|---|---|
| (1) WB-NRS 1 | - |  |  |  |  |  |  |  |  |
| (2) WB-NRS 2 | .70 | - |  |  |  |  |  |  |  |
| (3) WB-NRS 3 | .46 | .61 | - |  |  |  |  |  |  |
| (4) WB-NRS 4 | .54 | .62 | .64 | - |  |  |  |  |  |
| (5) WB-NRS 5 | .74 | .78 | .62 | .66 | - |  |  |  |  |
| (6) HADS-D | -.53 | -.57 | -.51 | -.47 | -.59 | - |  |  |  |
| (7) HADS-A | -.42 | -.49 | -.40 | -.38 | -.49 | .60 | - |  |  |
| (9) PSF-12 | .59 | .16# | .12# | .20# | .39* | -.41 | -.15# | - |  |
| (10) MSF-12 | .59 | .64 | .62 | .53 | .70 | -.70 | -.58 | .42 | - |

*Note*. N = 356. HADS-D = Hospital Anxiety and Depression Scale—Depression; HADS-A = Hospital Anxiety and Depression Scale—Anxiety; SF-12 = Short Form 12 items Health Survey. PSF-12 = Physical Health Index; MSF-12 = Mental Health Index.

#*ns*,

*$BF_{10} > 10$, for all the other correlations $BF_{10} > 100$.

**Table 4. Bivariate Bayesian correlates between the *Well-being Numerical Rating Scales* (WB-NRSs) and the other variables in the study in the non-clinical samples.**

| | (1) | (2) | (3) | (4) | (5) | (6) | (7) | (8) | (9) | (10) |
|---|---|---|---|---|---|---|---|---|---|---|
| (1) WB-NRS 1 | - | .38 | .29 | .38 | .45 | .39 | -.36 | .19[#] | .25 | .29 |
| (2) WB-NRS 2 | .57 | - | .60 | .61 | .77 | .62 | -.62 | .48 | .45 | .55 |
| (3) WB-NRS 3 | .47 | .61 | - | .47 | .67 | .56 | -.49 | .39 | .40 | .46 |
| (4) WB-NRS 4 | .32 | .49 | .41 | - | .61 | .51 | -.48 | .31 | .30 | .43 |
| (5) WB-NRS 5 | .59 | .73 | .65 | .52 | - | .70 | -.68 | .52 | .50 | .59 |
| (6) WHO-5 | .45 | .60 | .54 | .41 | .64 | - | -.68 | .53 | .53 | .61 |
| (7) PSS-4 | -.40 | -.64 | -.45 | -.41 | -.64 | -.57 | - | -.63 | -.53 | -.66 |
| (8) PSMS | .25 | .45 | .32 | .29 | .41 | .43 | -.63 | - | .59 | .71 |
| (9) LOT-R | .30 | .44 | .28 | .32 | .43 | .43 | -.45 | .50 | - | .61 |
| (10) SOC | .27 | .50 | .32 | .31 | .47 | .40 | -.51 | .53 | .37 | - |

*Note*. Above Diagonal = Sample 1 ($N$ = 231); Below Diagonal = Sample 2 ($N$ = 342). WHO-5 = World Health Organization Well-Being Index; PSS-4 = Perceived Stress Scale; PSMS = Pearlin-Schooler Mastery Scale; LOT-R = Life Orientation Test Revised; SOC = Sense of Coherence (SOC-13 item in Sample 1 and SOC-3 item in Sample 2).

[#]*ns*, *$BF_{10}$>10, for all the other correlations $BF_{10}$>100.

large in size correlations were observed in the clinical sample for depression (-0.57 < $r$ < -0.47). Comparable correlations were found for anxiety in the clinical sample (-0.49 < $r$ < -0.38) and for stress in the non-clinical samples (Sample 1: -0.64 < $r$ < -0.40; Sample 2: -0.68 < $r$ < -0.36). Positive and low to medium in size correlations were found for sense of mastery (0.19 < $r$ < 0.52 and 0.25 < $r$ < 0.45 in Sample 1 and Sample 2, respectively), dispositional optimism (0.25 < $r$ < 0.50 and 0.28 < $r$ < 0.43, respectively) and sense of coherence (0.29 < $r$ < 0.59 and 0.27 < $r$ < 0.47, respectively). Specifically, WB-NRS 1 was weakly correlated with these three individual dispositions or general resources, while the stronger correlations were observed for WB-NRS 2 and WB-NRS 5.

Finally, observing the pattern of correlations among the WB-NRSs and the PWBS six dimensions (Table 5), we found that that the WB-NRS 1 was not significantly correlated with *Autonomy*, *Personal Growth*, *Purpose in Life*, and *Positive Relations* (0.09 < $r$ < 0.14), and weakly correlated with *Self-acceptance* and *Environmental Mastery* ($r$ = 0.22 and $r$ = 0.28, respectively). The WB-NRS 4 showed low correlations with *Autonomy*, *Personal Growth*, *Purpose in Life*, and *Positive Relations* (0.21 < $r$ < 0.27), and medium correlations with *Self-acceptance* and *Environmental Mastery* ($r$ = 0.39 and $r$ = 0.38, respectively). Conversely, higher

**Table 5. Bivariate Bayesian correlates between the *Well-being Numerical Rating Scales* (WB-NRSs) and the *Psychological Well-being Scales* (PWBS).**

| | PWBS | | | | | |
|---|---|---|---|---|---|---|
| | *Self-acceptance* | *Autonomy* | *Environmental Mastery* | *Personal Growth* | *Purpose in Life* | *Positive Relations* |
| WB-NRS 1 | .22** | .12[#] | .28 | .14[#] | .12[#] | .09[#] |
| WB-NRS 2 | .53 | .27 | .53 | .28 | .38 | .29 |
| WB-NRS 3 | .47 | .32 | .48 | .34 | .38 | .57 |
| WB-NRS 4 | .39 | .21* | .38 | .21* | .27 | .21* |
| WB-NRS 5 | .57 | .31 | .57 | .31 | .46 | .36 |

*Note*. $N$ = 231 (Sample 1).

[#]*ns*,

*$BF_{10}$>10,

**$BF_{10}$>30, for all the other correlations $BF_{10}$>100.

correlation values were obtained for the remaining WB-NRS scales, but again smaller values were observed for *Autonomy*, *Personal Growth*, *Purpose in Life*, and *Positive Relations* ($0.27 < r < 0.36$, with the exception of the abovementioned strong relationship between WB-NRS 3 and *Positive Relations*) and higher values for *Self-acceptance* and *Environmental Mastery* ($0.48 < r < 0.57$).

By and large, the pattern of correlations supports the idea that the WB-NRS 5, which was correlated to all the measured constructs, captures a global well-being that encompasses different components, and it is markedly associated with the general resources linked to well-being (i.e., optimism, sense of mastery, and sense of coherence) as well as to other mental health indicators (i.e., depression, anxiety, and stress). However, the patterns of correlations observed for the WB-NRS 2 and WB-NRS 5 were quite similar, suggesting that there is a large overlap between the psychological and the general well-being. The WB-NRS 3 seems to assess a quite distinct well-being component. Indeed, when compared to the WB-NRS 2 and WB-NRS 5, it is similarly connected with some mental health indices and not correlated with the physical one, but the relationships with general resources, stress, and anxiety are weaker. Finally, observing the correlations, the WB-NRS 1 and WB-NRS 4 appears to tap different and peculiar aspects of well-being.

**Responsiveness.** The WB-NRSs were used to detect the change in well-being in cancer patients who were invited to attended live classical music concerts during the hospitalization period. The one-sided $t$-test showed that the resulting $BF_{10}$ value was 9.11 for WB-NRS 5, indicating substantial evidence in favor of alternative hypothesis, i.e., general well-being increased after the intervention. The same results were obtained for WB-NRS 1 ($BF_{10} = 5.35$), suggesting a specific effect on physical well-being. A small effect was detected for WB-NRS 2 ($BF_{10} = 3.03$), showing a minimal change in psychological well-being. The $BF_{10}$ values were 0.11 and 0.15 for WB-NRS 3 WB-NRS 4, respectively, indicating there were not pre-post differences in relational and spiritual well-being.

The WB-NRSs were also administered to detect changes in well-being induced by the immersion in an enriched environment in a museum in Florence. The one-sided $t$-test showed that the resulting $BF_{10}$ value was 113.44 for WB-NRS 5, indicating very strong evidence in favor of alternative hypothesis, i.e., subjective general well-being dramatically increased after the intervention. A similar result was obtained for WB-NRS 2 ($BF_{10} = 32.80$), suggesting a specific large effect on psychological well-being. The $BF_{10}$ value was 0.29, 2.03, and 2.14 for WB-NRS 1, WB-NRS 3, and WB-NRS 4, respectively, showing there were not pre-post differences in physical, relational, and spiritual well-being.

Overall, the responsivity results attested that the WB-NRSs were able to detect changes in well-being after an intervention program and that a specific change can be detected. In the first case, a desirable effect was to reduce patients' physical distress related to treatments and to increase their psychological and general well-being. The WB-NRSs allowed detecting these specific effects. Similarly, the second intervention aimed basically to increase psychological and general well-being. Also, in this case the WB-NRSs were able to show these changes. Conversely, changes in relational and spiritual well-being were not an expected outcome in these interventions, and the results confirmed this hypothesis.

## Study 2

The purpose of the current study was twofold: to test the metrical equivalence of the Italian and the English version of the WB-NRSs and to provide evidence of validity confirming the pattern of relationships observed in Study 1 and replicating the same pattern in the English sample.

## Methods

**Participants.** The English-speaking sample consisted of 287 Canadian undergraduate psychology students (ages ranged from 18 to 30 years; *M* = 20.78, *SD* = 3.32; 23% males) from a large Canadian university. The Italian-speaking sample consisted of 342 Italian undergraduate psychology students (age ranged from 18 to 29 years, *M* = 21.21 years, *SD* = 1.73, 38% males). Participation in the study was voluntary and students received a credit towards their psychology course. The study was approved by the university's local institutional review boards (Commissione Etica per la Ricerca dell'Università degli Studi di Firenze, n. 31—prot. 127556 and Western University Non-Medical Research Ethics Board–NMREB Project ID: 116791, n. 00000941). Informed consent was obtained from all participants.

**Design, measures, and procedure.** A descriptive observational research design was adopted; specifically, it was a cross-sectional study.

The five items of the WB-NRSs were translated by two Italian psychologists and the translated version was edited by an English-speaking researcher. Since a NRS requires minimal language translation, no further steps in the translation process were deemed necessary. As described above (Study 1), respondents are asked to rate their physical (WB-NRS 1), psychological (WB-NRS 2), relational (WB-NRS 3), spiritual (WB-NRS 4), and general well-being (WB-NRS 5) using a 1 to 10 scale with 1 meaning a state of "absolute distress" and 10 a state of "complete well-being."

The WB-NRSs was presented to each sample as part of an online questionnaire (Qualtrics was used for Canadian participants; GoogleForms for Italian participants) including the following scales (see Study 1 for a detailed description): the LOT-R (the scale has a good internal consistency in the current Canadian and Italian samples: $\omega$ = .86 and $\omega$ = .84, respectively); the PSMS (Canadian sample: $\omega$ = .85; Italian sample: $\omega$ = .80), the WHO-5 ($\omega$ = .84 and .81, respectively), and the PSS-4 ($\omega$ were .81 and .78, respectively, indicating adequate reliability).

**Analysis strategy.** Missing values treatment and preliminary analyses to IRT were the same of Study 1.

To investigate measurement equivalence, an analysis of differential item functioning (DIF) across groups was performed by applying the Item Response Theory Likelihood Ratio test approach (IRTLR) [57] implemented in IRTPRO software 4.0 [42]. The analysis refers to IRT *b* (threshold) and *a* (discrimination) parameter estimates. Hence, two basic types of DIF are examined. Uniform DIF refers to threshold parameters and non-uniform DIF refers to discrimination parameters. The program uses a chi-square statistic to evaluate the difference of parameters across the groups. Given the multiple comparisons, the critical *p*-value was adjusted at .01 (i.e., *p*-value < .01 can be interpreted as indicators of DIF).

Bayesian correlation tests were conducted to test the validity of the WB-NRSs (see Study 1).

## Results and discussion

Minimal data were missing across all variables. For the WB-NRSs, the missing values were about 1% of the total cases in the samples. Thus, listwise deletion was used and 4 cases were excluded.

**Prerequisites for IRT analysis.** *Item descriptives*. All indices are reported in Table 6. Descriptives of each item showed that respondents used all the possible scale options with the exception of the WB-NRS 1 where the option 1 was never chosen in both groups. Answers were well distributed along the range of the numerical scale (standard deviations were all around 2). Means (values ranged from 6.2 to 7.2) and skewness indices (all negative in sign) indicated that the frequencies were slightly fewer in the lower part of the response scale (i.e., answers were more concentrated on the higher values). Nonetheless, all the skewness and

**Table 6. Descriptives, item-total correlations (with item deleted), and factor loadings of the *Well-being Numerical Rating Scales* (WB-NRSs) in the Canadian and Italian samples.**

| Sample | Item | Range | Mean | Standard Deviation | Skewness | Kurtosis | Item-Total correlation | Factor Loading |
|--------|------|-------|------|--------------------|----------|----------|------------------------|----------------|
| Canadian | WB-NRS 1 | 2–10 | 7.23 | 1.65 | -0.72 | 0.57 | 0.54 | 0.62 |
| | WB-NRS 2 | 1–10 | 6.15 | 2.11 | -0.29 | -0.72 | 0.76 | 0.85 |
| | WB-NRS 3 | 1–10 | 6.93 | 1.98 | -0.76 | 0.43 | 0.59 | 0.67 |
| | WB-NRS 4 | 1–10 | 6.58 | 2.20 | -0.46 | -0.41 | 0.60 | 0.66 |
| | WB-NRS 5 | 1–10 | 6.92 | 1.70 | -0.64 | 0.37 | 0.87 | 0.98 |
| Italian | WB-NRS 1 | 2–10 | 6.75 | 1.79 | -0.51 | -0.09 | 0.49 | 0.55 |
| | WB-NRS 2 | 1–10 | 6.29 | 1.95 | -0.56 | -0.37 | 0.76 | 0.84 |
| | WB-NRS 3 | 1–10 | 6.99 | 2.03 | -0.87 | 0.31 | 0.70 | 0.77 |
| | WB-NRS 4 | 1–10 | 6.17 | 2.24 | -0.36 | -0.37 | 0.63 | 0.69 |
| | WB-NRS 5 | 1–10 | 6.75 | 1.75 | -0.85 | 0.79 | 0.87 | 0.97 |

kurtosis indices ranged within -1 and 1, suggesting there is not a substantial departure from a normal distribution. Item–total correlation values ranged from .59–.81 suggesting that the variability in item discrimination parameters would warrant inclusion in the IRT calibration.

*Internal Structure.* The parallel analysis clearly advised a one-factor structure in both samples. The model showed an excellent fit in the Canadian ($\chi^2/df$ = 1.62, CFI = .997, NNFI = .993, RMSEA = .047 [95%CI: .005 -.078]) and Italian data ($\chi^2/df$ = 0.45, CFI = .999, NNFI = .999, RMSEA = .001 [95%CI: .000 -.070]). The variance explained by the model was 65% and 67% in the Canadian and Italian sample, respectively. Factor loadings were higher than 0.60 in the Canadian sample and higher than 0.50 in the Italian one (Table 6).

*Local Dependence.* In both samples, none of the LD statistics were greater than 10 (the maximum value was 6.4), indicating the absence of covariations between couples of items that are not accounted for by a unidimensional model.

Overall, these results showed that DIF analysis can be performed to describe the psychometric properties of the items.

**IRT differential item functioning.** The DIF analyses in which the Canadian group was the reference group, indicated that WB-NRS 1, WB-NRS 2, WB-NRS 3, and WB-NRS 4 showed neither uniform DIF ($X^2$ statistics ranged from 7.8 to 12.5, with associated *p*-values ranging from .56 to .13), nor non-uniform DIF ($X^2$ statistics ranged from 0.0 to 2.8, with associated *p*-values ranging from .86 to .09). The WB-NRS 5 did not showed DIF for the *b* parameters ($X^2(9)$ = 9.0, *p* = .44) but a difference was detected for the *a* parameter ($X^2(1)$ = 10.5, *p* <

**Table 7. Differential item functioning analyses of the *Well-being Numerical Rating Scales* (WB-NRSs) across Italian and Canadian samples.**

| Item | Total | | | a | | | b | | |
|------|-------|------|------|------|------|------|------|------|------|
| | $X^2$ | df | p | $X^2_a$ | df | p | $X^2$ | df | p |
| WB-NRS 1 | 13.7 | 9 | 0.131 | 1.3 | 1 | 0.260 | 12.5 | 8 | 0.131 |
| WB-NRS 2 | 12.5 | 10 | 0.252 | 0.0 | 1 | 0.861 | 12.5 | 9 | 0.187 |
| WB-NRS 3 | 10.6 | 10 | 0.392 | 2.8 | 1 | 0.094 | 7.8 | 9 | 0.558 |
| WB-NRS 4 | 8.6 | 10 | 0.567 | 0.6 | 1 | 0.438 | 8.0 | 9 | 0.531 |
| WB-NRS 5 | 19.4 | 10 | 0.033 | 10.5 | 1 | 0.001 | 9.0 | 9 | 0.440 |

*Note.* DIF was calculate under Samejima's Graded Response Model. α was considered significant at .01 (0.05/5) in order to adjust for multiple comparisons (Bonferroni correction). *a* = discrimination, *b* = category threshold.

.001). Nonetheless, the $X^2$ total statistics indicated that all the WB-NRSs can be considered invariant across language (Table 7).

**Validity.** Bayesian correlations are reported in Table 8. As a preliminary step to analyses, we checked the normality of the score distributions. All the skewness and kurtosis indices ranged within -1 and 1, suggesting that distributions were similar to a normal distribution.

As expected, high correlations were observed between comparable dimensions of well-being, such as the WB-NRS 2 and the WHO-5 ($r = 0.71$ and $r = 0.61$ in the Canadian and Italian sample, respectively). Low to medium correlations were observed between different dimensions of well-being, such as the WB-NRS 1 and the WHO-5 ($r = 0.47$ and $r = 0.38$, in the Canadian and Italian sample, respectively). The WB-NRS 4 was moderately correlated with WHO-5 ($r = 0.50$ and $r = 0.47$ in the Canadian and Italian sample, respectively). Finally, consistently with the proposed theoretical model, WB-NRS 5 was strongly correlated with the WHO-5 ($r = 0.70$ and $r = 0.66$ in the Canadian and Italian sample, respectively). These results confirm the ones obtained in Study 1 and attest that the observed relationships can be replicated using the English version of the WB-NRSs.

Additionally, negative and medium to large in size correlations were observed for stress (Canadian: $-0.65 < r < -0.38$; Italian: $-0.66 < r < -0.29$). Positive and low to medium in size correlations were found for sense of mastery ($0.27 < r < 0.49$ and $0.28 < r < 0.50$ in the Canadian and Italian sample, respectively) and dispositional optimism ($0.25 < r < 0.50$ and $0.28 < r < 0.43$, respectively). By and large, the lower correlations were observed among the WB-NRS 1 and these individual dispositions, while the stronger correlations were observed for WB-NRS 2 and WB-NRS 5. Also, these results were very similar to the ones obtained in Study 1 and that the observed relationships were replicated for the English version of the WB-NRSs.

## General discussion

The present study aimed to develop a reliable and valid tool to assess well-being fitting together a multidimensional approach to well-being and the added value offered by a brief scale (e.g., reduction of respondents' fatigue, boredom, and loss of interest, suitability for large, multivariate surveys with many tests and scales, rapid assessment in pre-post administration). To achieve this goal, we deemed that the numerical rating scale as an optimal solution because it is brief, concise, and easy to understand, interpret, and score. Thus, relying on the theoretical model represented in Fig 1, we developed the *Well-being Numerical Rating Scales* (WB-NRSs) consisting of five NRSs used to rate physical, psychological, spiritual, relational, and general well-being.

**Table 8. Bivariate Bayesian correlates between the *Well-being Numerical Rating Scales* (WB-NRSs) and the other variables in the study.**

|  | **(1)** | **(2)** | **(3)** | **(4)** | **(5)** | **(6)** | **(7)** | **(8)** | **(9)** |
|---|---|---|---|---|---|---|---|---|---|
| (1) WB-NRS 1 | - | .50 | .33 | .40 | .60 | .47 | -.38 | .27 | .29 |
| (2) WB-NRS 2 | .42 | - | .53 | .56 | .82 | .71 | -.65 | .49 | .54 |
| (3) WB-NRS 3 | .39 | .65 | - | .44 | .67 | .50 | -.43 | .37 | .40 |
| (4) WB-NRS 4 | .37 | .58 | .51 | - | .58 | .50 | -.48 | .34 | .32 |
| (5) WB-NRS 5 | .53 | .81 | .74 | .65 | - | .70 | -.63 | .49 | .51 |
| (6) WHO-5 | .38 | .61 | .49 | .47 | .66 | - | -.66 | .50 | .55 |
| (7) PSS-4 | -.29 | -.66 | -.47 | -.46 | -.66 | -.63 | - | -.70 | -.63 |
| (8) PSMS | .28 | .50 | .42 | .41 | .49 | .46 | -.56 | - | .66 |
| (9) LOT-R | .29 | .39 | .34 | .37 | .43 | .47 | -.51 | .50 | - |

*Note*. Below Diagonal = Italian sample ($N = 342$); above Diagonal = Canadian sample ($N = 283$). WHO-5 = World Health Organization Well-Being Index; PSS-4 = Perceived Stress Scale; PSMS = Pearlin-Schooler Mastery Scale; LOT-R = Life Orientation Test Revised. $BF_{10} > 100$ for all the correlations.

Administering the scales to a large sample consisting of clinical and non-clinical respondents, we were able to provide evidence that the developed tool has adequate psychometric properties. Through applying IRT, we showed that each NRS performs adequately in measuring the targeted well-being component. Specifically, each item has a good discriminant ability, and the spread of threshold parameters attests the appropriateness of the response categories. Along with item properties, the pattern of correlations with other measures of general, psychological, relational, existential, and physical well-being confirmed that each scale assesses a specific dimension of well-being and that overall well-being is not simply the summation of all these components. Moreover, we were able to replicate the nomological net documented in the literature investigating the relationships of well-being with dispositional optimism, sense of coherence, sense of mastery, stress, anxiety, and depression [1]. Finally, we observed that the WB-NRSs were able to assess changes due to intervention programs developed in the health domain [23, 24] and in the art domain [25, 26]. Specifically, the responsivity results attested that the WB-NRSs were able to detect the specific well-being changes consistently with the aim of the proposed intervention.

Interestingly, findings with the threshold parameters suggested that the WB-NRS assessed individuals at the lower end of the well-being (i.e., one or two standard deviations below the mean) as well as the higher end of well-being. Typically, the latent trait of interest is a quasi-trait, which is a unipolar construct measuring presence or absence of a trait (e.g., content vs. non-content [58]). This is in contrast with a bipolar trait, in which both extremes on opposite ends represent variations in two meaningful entities (e.g., content vs. distress). Reise and Waller suggested that this discrepancy can be explained as the low end of a quasi-trait is not a meaningful construct, and as such, there is no need to measure the extreme end of the spectrum in a quasi-trait [58]. It appears that the WB-NRS may act as a bipolar trait scale of measuring distress on one end and well-being on another. One limitation of many personality variable scales may be that of any quasi-trait scale, such that evaluating change may be especially difficult for individuals where only one end of the measure assesses a meaningful entity [58]. As such, the WB-NRSs may address this important limitation.

Previous findings have demonstrated item-total scores tended to be positively skewed and threshold parameters tend to be clustered when assessing clinical samples [58]. Individuals two standard deviations below the mean highly likely to endorse specific responses on one end of the continuum [58, 59]. The present findings suggested threshold parameters in this measure spread across the latent continuum, which may reflect a larger sample utilized for this study and/or items are well-spread across the continuum to adequately capture the trait.

Although a NRS requires minimal language translation, we conducted a second study to confirm the possibility to use the WB-NRSs once translated in English. An IRT-based Differential Item Functioning analysis provided evidence that the item properties are similar for the Italian and English version of the scale. As such, the WB-NRSs function equivalently regardless the respondent's language. Additionally, in this second study, we confirmed the validity results obtained in Study 1 and we observed very similar relationships when comparing the Italian and the English-speaking samples.

While the two studies provide a broader support for the use of the WB-NRSs for a rapid and sound assessment of well-being, there are some limitations and, consequently, suggestions for future research. First of all, two administration formats (pencil-and-paper and online surveys) were used in the clinical and non-clinical samples This methodological dissimilarity should be avoided in future investigations to confirm and strengthen the current results. Secondly, to test invariance across language we employed samples of university students. Thus, further studies are needed to generalize the usability of the scale to the English-speaking general and clinical population akin to the Italian samples. Finally, validity testing might be extended including other measures of the different well-being dimensions. In particular, a

spiritual well-being measure should be included to better understand the ability of the current NRS to assess this specific well-being component.

Overall, the current study provides evidence that the Italian and English versions of the WB-NRSs offer added value in research focused on well-being and in assessing well-being changes prompted by intervention programs developed for clinical and non-clinical purposes.

## Supporting information

**S1 File. WB Study 1.** Study 1: Well-being datafile.
(TXT)

**S2 File. WB Study 1 validity clin.** Study 1: Well-being validity datafile (clinical sample).
(TXT)

**S3 File. WB Study 1 validity no clin 1.** Study 1: Well-being validity datafile (non-clinical sample 1).
(TXT)

**S4 File. WB Study 1 validity no clin 2.** Study 1: Well-being validity datafile (non-clinical sample 2).
(TXT)

**S5 File. WB Study 1 pre-post clin.** Study 1: Well-being pre-post datafile (clinical sample).
(TXT)

**S6 File. WB Study 1 pre-post no clin.** Study 1: Well-being pre-post datafile (non-clinical sample).
(TXT)

**S7 File. WB Study 2.** Study 2: Well-being datafile.
(TXT)

**S8 File. WB Study 2 validity Italy.** Study 2: Well-being validity datafile (Italian sample).
(TXT)

**S9 File. WB Study 2 validity Canada.** Study 2: Well-being validity datafile (Canadian sample).
(TXT)

**S1 Appendix.**
(DOCX)

## Author Contributions

**Conceptualization:** Andrea Bonacchi, Francesca Chiesi.

**Data curation:** Andrea Bonacchi, Francesca Chiesi, Chloe Lau, Georgia Marunic.

**Formal analysis:** Francesca Chiesi.

**Investigation:** Andrea Bonacchi, Chloe Lau, Georgia Marunic, Fabio Marra, Guido Miccinesi.

**Methodology:** Francesca Chiesi.

**Supervision:** Andrea Bonacchi, Donald H. Saklofske, Fabio Marra, Guido Miccinesi.

**Validation:** Francesca Chiesi.

**Writing – original draft:** Andrea Bonacchi, Francesca Chiesi, Georgia Marunic.

**Writing – review & editing:** Chloe Lau, Donald H. Saklofske.

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
