## [Decision Letter · Decision Letter 0]

7 Apr 2021

PONE-D-21-01391

Rapid and sound assessment of well-being within a multi-dimensional approach: The Well-being Numerical Rating Scales (WB-NRSs)

PLOS ONE

Dear Dr. Chiesi,

Thank you for submitting your manuscript to PLOS ONE. After careful consideration, we feel that it has merit but does not fully meet PLOS ONE’s publication criteria as it currently stands. Therefore, we invite you to submit a revised version of the manuscript that addresses the points raised during the review process.

We look forward to receiving your revised manuscript.

Kind regards,

Paola Gremigni, Ph.D.

Academic Editor

PLOS ONE

Journal Requirements:

3. We noted in your submission details that a portion of your manuscript may have been presented or published elsewhere:

'To test responsiveness, we employed data that were partially published in a conference proceeding and in a paper. We re-used these pre-post data in a different perspective to provide evidence of the psychometric properties of the scale, i.e., the ability to detect changes.'

Please clarify whether this conference proceeding  was peer-reviewed and formally published.

If this work was previously peer-reviewed and published, in the cover letter please provide the reason that this work does not constitute dual publication and should be included in the current manuscript.

Reviewers' comments:

Reviewer's Responses to Questions

**Comments to the Author**

1. Is the manuscript technically sound, and do the data support the conclusions?

Reviewer #1: Yes

Reviewer #2: Yes

2. Has the statistical analysis been performed appropriately and rigorously? 

Reviewer #1: Yes

Reviewer #2: Yes

3. Have the authors made all data underlying the findings in their manuscript fully available?

Reviewer #1: Yes

Reviewer #2: Yes

4. Is the manuscript presented in an intelligible fashion and written in standard English?

Reviewer #1: Yes

Reviewer #2: Yes

5. Review Comments to the Author

Reviewer #1: This review is intended to indicate some minor modifications listed below

1 – rows 99 and 114. Two different goals are presented. It is advisable to unify the presentation of the aim.

2 – row 129 – I would suggest clarifying that the proposed interventions are based on the literature, which attests to their effectiveness. Such reinforcement would be a clarification indicating the effectiveness of the proposed measure, as the intervention must bring about such a change.

3 – In study 1 – methods - clarify the use of such a heterogeneous sample. And how this choice of sample could be positive for the study.

4 – row 176 - who was the convenience group? Does it represent the cognitive capacity of the target sample of the Scale?

5 – In study 1 - Design and procedure - indicate when it was carried out the data collection. Having been in the period of the pandemic, if there was a need to change the form of the collection.

6 - Whereas the scales have different scores. Was a procedure used to standardize the score to ensure compatibility between the measurements? Especially in the discriminant validity?

7 - Could the absence of a common measure (in addition to the WB-NRS) in the sample subgroups hinder the analysis of discriminant validities?

8 - Has the normality of the distribution of data for statistical tests that require this premise been guaranteed?

9 – Study 2 – row 532 - indicate when it was carried out the data collection, and the collection server (e.g. googleforms, surveymonkey…)

10 - It is suggested to update the references.

Reviewer #2: Dear respectful

Prof. Paola Gremigni

Thank you for choosing me as reviewer for paper entitled “Rapid and sound assessment of well-being within a multi-dimensional approach: The Well-being Numerical Rating Scales (WB-NRSs).

In brief, Overall, the current study provides evidence that the Italian and English versions of the

WB-NRSs offer added value in research focused on well-being and in assessing well-

being changes prompted by intervention programs.

As you see my work as reviewer for Plose one, evaluating many articles, I accepted some of them with modification. I rejected some of them. Regarding the current paper, I recommend this paper as best paper for 2021 in your esteemed journal from point of empirical paper.

Major advantages and merits of paper as follow as

1)Authorities used nine scales to validate the established wellbeing. We cannot find this number of scales in past studies. Even if there, it is few.

2)Two studies are presented in one paper.

Clinical sample and non-clinical sample (students with their families).

Canadian and Italian samples.

3) Intervention based on Music

4) Scare and modern statistics have been precisely used in paper. Such as MacDonald reliability, Bayesian correlation, parallel analysis, item response theory (differential item functioning). Obtained results were in consistent with logic, theory, two samples and past studies. I am familiar with all kinds of statistics.

5) Written Language of paper is accurately scientific understandable

6) High concentration in presentation of theories as well as discussion.

Finally I recommended this research team to be continuous for doing researches together (cross field –epidemiology and psychology= cross culture –Canadian and Italy).

Minor points

I accepted the article without modification. However,

Demographic variables (countries) are not presented in Study one, why?

Factor loadings for Canadian are higher than Italian? Please, give interpretation for this result?

Factor loadings discovered difference between two nations, while differential item functioning confirmed similarities. Please why results are discrepant? Please interpret?

Best regards

DR. Nasser Alareqe

Malaysia

6. PLOS authors have the option to publish the peer review history of their article (what does this mean?). If published, this will include your full peer review and any attached files.

Reviewer #1: **Yes: **Victor Zaia

Reviewer #2: **Yes: **DR. Nasser Alareqe Malaysia

---

## [Author Response · Author response to Decision Letter 0]

18 May 2021

Reviewer #1: 

Comment 1: Two different goals are presented. It is advisable to unify the presentation of the aim.

Response: Thank you to the reviewer for pointing this out. We have modified accordingly.

Comment 2: I would suggest clarifying that the proposed interventions are based on the literature, which attests to their effectiveness. Such reinforcement would be a clarification indicating the effectiveness of the proposed measure, as the intervention must bring about such a change.

Response: Thank you to the reviewer for asking for further clarification. We specified that both the interventions were original strategies (using music and artistic features). Thus, there is not a specific literature to refer to. Provided that the intervention’s aim was to improve participants well-being, the effectiveness of the proposed measure in detecting such a change can be considered an important feature of the instrument.

Comment 3: In Study 1 clarify the use of such a heterogeneous sample. And how this choice of sample could be positive for the study.

Response: Thank you to the reviewer for asking for further clarification. We specified that the sample allowed us to have a representative sample of the population by age and gender, and that including clinical participants allowed us to test the effectiveness of the scale in the health domain where it is especially important consider psychosocial well-being while treating disease. As such, the scale that was developed is not only reliable and valid, but may also be generalizable to a variety of age groups and clinical and general population settings. 

Comment 4: Who was the convenience group? Does it represent the cognitive capacity of the target sample of the Scale?

Response: Thank you to the reviewer for this suggestion. The cognitive capacity of the students were not assessed as part of data collection. The sample was a convenience sample because they were university students part of a psychology course. Provided that they were admitted into university, the authors could speculate that they were of normal or above average in terms of cognitive capacity compared to the general population. We also believe that the task at hand (i.e., filling out a questionnaire where the well-being measure is a short descriptor) would be manageable for a wide range of individuals of different ranges of cognitive capacity and reading levels. 

Comment 5: In study 1 - Design and procedure - indicate when it was carried out the data collection. Having been in the period of the pandemic, if there was a need to change the form of the collection.

Response: Thank you to the reviewer for asking this clarification. Data were collected before the pandemic, specifically from January 2017 to April 2019.

Comment 6: Whereas the scales have different scores. Was a procedure used to standardize the score to ensure compatibility between the measurements? Especially in the discriminant validity? Response: Thank you to the reviewer for allowing to clarify this point. Since we performed correlational analyses, standardization was not required given that the correlation is normalized by standard deviation.

Comment 7: Could the absence of a common measure (in addition to the WB-NRS) in the sample subgroups hinder the analysis of discriminant validities? 

Response: Thank you to the reviewer for allowing to clarify this point. As stated in Study 1 Analysis Strategy, Discriminant validity was tested to demonstrating that a measure does not correlate too strongly with measures they are not intended to. Thus, correlations between couples of variables were performed. We thank you for this suggestion that might be tested in future studies.

Comment 8: Has the normality of the distribution of data for statistical tests that require this premise been guaranteed? 

Response: Thank you to the reviewer for asking for this specification. The W-BNRSs scores were normally distributed (see kurtosis and skewness indices in Table 1 and Table 6). Now, we added that these indices fall within this range for all the variables in the study.

Comment 9: Study 2 – row 532 - indicate when it was carried out the data collection, and the collection server (e.g. googleforms, surveymonkey…)

Response: Thank you to the reviewer for asking for this specification. We added the names of the servers.

Comment 10 - It is suggested to update the references.

Response: References were updated.

Reviewer #2: 

First of all, we would like to thank you and express our gratitude for the appreciation of our work.

Comment: Demographic variables (countries) are not presented in Study one, why? 

Response: Thank you to the reviewer for pointing this out. Study 1 was conducted with Italian participants.

Comment: Factor loadings for Canadian are higher than Italian? Please, give interpretation for this result? Factor loadings discovered difference between two nations, while differential item functioning confirmed similarities. Please why results are discrepant? Please interpret? 

Response: Thank you to the reviewer for pointing this out. Following this suggestion, we conducted a multigroup CFA. Results indicated that Metric invariance holds, i.e., measurement weights did not lead to a significant decrement in model fit compared with the configural model when constrained to be equal (ΔChi2= 6.91, p=.141; ΔCFI = .002; ΔRMSEA= .005). These results suggest metric invariance can be established and in line with IRT results, the scale is invariant across these two groups. Given that IRT does cover differential item functioning, we prefer not to add further analyses beca

---

## [Editor Report · Decision Letter 1]

21 May 2021

Rapid and sound assessment of well-being within a multi-dimensional approach: The Well-being Numerical Rating Scales (WB-NRSs)

PONE-D-21-01391R1

Dear Prof. Chiesi,

We’re pleased to inform you that your manuscript has been judged scientifically suitable for publication and will be formally accepted for publication once it meets all outstanding technical requirements. This Academic Editor also appreciated your competent answers to all the Reviewers' suggestions.

Kind regards,

Prof. Paola Gremigni, Ph.D.

Academic Editor

PLOS ONE
---

## [Editor Report · Acceptance letter]

27 May 2021

PONE-D-21-01391R1 

Rapid and sound assessment of well-being within a multi-dimensional approach: The *Well-being Numerical Rating Scales* (WB-NRSs) 

Dear Dr. Chiesi:

I'm pleased to inform you that your manuscript has been deemed suitable for publication in PLOS ONE. Congratulations! Your manuscript is now with our production department. 

Kind regards, 

on behalf of

Prof. Paola Gremigni 

Academic Editor

PLOS ONE